# Resolution of the ordinal phylogeny of mosses using targeted exons from organellar and nuclear genomes

Yang Liu[1,2], Matthew G. Johnson[3], Cymon J. Cox[4], Rafael Medina[5], Nicolas Devos[6], Alain Vanderpoorten[7], Lars Hedenäs[8], Neil E. Bell [9], James R. Shevock[10], Blanka Aguero[6], Dietmar Quandt[11], Norman J. Wickett[12], A. Jonathan Shaw[6] & Bernard Goffinet [13]

Mosses are a highly diverse lineage of land plants, whose diversification, spanning at least 400 million years, remains phylogenetically ambiguous due to the lack of fossils, massive early extinctions, late radiations, limited morphological variation, and conflicting signal among previously used markers. Here, we present phylogenetic reconstructions based on complete organellar exomes and a comparable set of nuclear genes for this major lineage of land plants. Our analysis of 142 species representing 29 of the 30 moss orders reveals that relative average rates of non-synonymous substitutions in nuclear versus plastid genes are much higher in mosses than in seed plants, consistent with the emerging concept of evolutionary dynamism in mosses. Our results highlight the evolutionary significance of taxa with reduced morphologies, shed light on the relative tempo and mechanisms underlying major cladogenic events, and suggest hypotheses for the relationships and delineation of moss orders.

[1] Fairy Lake Botanical Garden & Chinese Academy of Sciences, Shenzhen 518004, China. [2] BGI-Shenzhen, Shenzhen 518120, China. [3] Texas Tech University, Lubbock, TX 79409, USA. [4] Centro de Ciências do Mar, Universidade do Algarve, Gambelas, 8005-319 Faro, Portugal. [5] Department of Biology, Augustana College, Rock Island, IL 61201, USA. [6] Department of Biology, Duke University, Durham, NC 27708, USA. [7] Institute of Botany, University of Liège, Liège 4000, Belgium. [8] Department of Botany, Swedish Museum of Natural History, Stockholm Box 50007, 10405, Sweden. [9] Royal Botanic Garden Edinburgh, 20A Inverleith Row, Edinburgh EH3 5LR, UK. [10] California Academy of Sciences, San Francisco, CA 94118, USA. [11] Nees Institute for Biodiversity of Plants, University of Bonn, Bonn 53115, Germany. [12] Chicago Botanic Garden, Glencoe, IL 60022, USA. [13] Department of Ecology and Evolutionary Biology, University of Connecticut, Storrs, CT 06269, USA. These authors contributed equally: Yang Liu, Matthew G. Johnson. Correspondence and requests for materials should be addressed to B.G. (email: bernard.goffinet@uconn.edu)

Bryophytes, a group of extant land plants, including mosses, liverworts, and hornworts, are characterized by a life cycle dominated by a vegetative gametophyte. Unlike extant vascular plants, which develop an independent and branched sporophyte, bryophytes permanently retain the sporophyte on the maternal gametophyte, and their sporophyte remains unbranched, and terminated by a single sporangium. The diversification of mosses, which may span at least 400[1–3], is marked by fundamental transformations of both generations of the life cycle[4]. In parallel to the multiple innovations to structures that regulate spore dispersal, the vegetative body has undergone developmental shifts optimizing vegetative growth while allowing for concurrent sexual reproduction[5,6]. These transformations, along with the general ability of mosses to withstand dehydration and desiccation[7], may account for their persistence and continued diversification since at least the Devonian. Today 13,000 species[6], distributed across all terrestrial biomes, contribute critical ecological functions[8] and play important roles in global biogeochemical cycles[9]. This diversity of mosses arose from repeated bursts of diversification[2] that may have been triggered by extrinsic factors such as changes in global climate[10], or intrinsic factors such as whole genome duplications[11]. The sequence of innovations in modes of spore dispersal and body growth remains, however, poorly understood due to a combination of factors, including massive early extinctions[2], late radiations[10], extreme paucity in Paleozoic fossil, limited levels of morphological variation, and phylogenetic uncertainty of the order of divergences among major moss groups that make up the ordinal relationships of the moss tree of life[4].

Transformations in the mechanisms that regulate spore dispersal most conspicuously mark the differences between the five major lineages of mosses (Fig. 1). The relationships between the Takakiophytina and Sphagnophytina, or between the Andreaeobryophytina and Andreaeophytina are incongruent among reconstructions[12–16]. Similarly, the sequence of events in the evolution of the Bryophytina remains ambiguous. This lineage comprises 90% of extant mosses and is characterized by the development of a peristome, comprising one or two rings of typically hygroscopic teeth that line the sporangium opening and regulate spore release[17]. Peristome types differ in their architecture and ontogeny and these diagnose major lineages[18] (Fig. 1). Homology among peristomial traits remains, however, ambiguous[6], including between the two basic peristome architectures, namely the nematodontous type wherein teeth are composed of whole cells, and the arthrodontous type, which consists of only partial plates of cell walls[19]. Reconstructing ancestral states is further hampered by uncertainty of relationships between nematodontous and arthrodontous mosses and the resolution of lineages lacking a peristome, in critical phylogenetic positions (e.g., Oedipodiopsida)[12,20]. Within arthrodontous mosses (i.e., Bryopsida), early splits gave rise to main lineages each defined by a unique peristome architecture (e.g., Timmiidae, Funariidae, Dicranidae, Bryidae). The relative position of these lineages is, however, incongruent among inferences[13,21,22] and ontogenetic features are too few, and still insufficiently sampled across the phylogeny, to offer robust insight in the succession of transformations[6]. Finally, the origin and diversification of the most species-rich lineage of Bryopsida, the pleurocarps or Hypnanae, which hold perhaps 50% of moss diversity, have also been contentious. This lineage is characterized by the development of female sex organs in a lateral and nearly sessile position on the stem terminating short branches (i.e., pleurocarpy) vs. at the apex of the stem (i.e., acrocarpy; Fig. 1). Pleurocarpy may be considered a key innovation as it enables continuous vegetative growth of the main module[5], and may have allowed for the subsequent rapid and extensive radiation of the Hypnanae[23].

A number of intermediate forms (called cladocarps or proto-pleurocarps) exist, although the precise distinction between these growth forms and pleurocarpy has been controversial[5,24]. All groups outside of the universally pleurocarpous Hypnanae are wholly or partially acrocarpous and the sister-group to the Hypnanae remains to be robustly identified. Similarly, the delineation of the major lineages of Hypnanae remains ambiguous in light of phylogenetic inferences based on few discrete loci[21,25]. This persistent phylogenetic ambiguity reflects the difficulty of reconstructing ancient splits or rapid diversifications from variation in a few discrete loci, and calls for the assembly of large datasets[26].

Phylogenomics, that is, phylogenetic inferences from a large set of genes, has contributed to the resolution of ancient and rapid divergences in several plant groups[27]. Many studies in plant phylogenomics rely on inferences from plastid genome data[28], which are easily obtained but collectively represent one (assumed) uni-parentally inherited phylogenetic history. By contrast, the mitochondrial genome is often ignored as a source of phylogenetic information, perhaps because of its slow rate of molecular evolution[29,30], which may, however, make it suitable for inferring deep relationships[31]. Transcriptomes have frequently provided the source of nuclear data for phylogenomic reconstruction of plants[11,32], but the requirement for fresh tissue limits their applicability. Alternatively, targeted enrichment methods provide an efficient and cost-effective alternative for the acquisition of extensive nuclear data from preserved samples, including of non-model organisms[33], and have been applied to phylogenetic reconstructions in animals[34,35] and plants[27]. One difficulty in applying targeted sequencing to reconstruct high-level relationships in plants is identifying genes with clear orthology and with sufficient numbers of variable sites. Non-coding regions are most commonly targeted[34,35], but these may lead to ambiguities in the alignment due to the difficulty of assessing homology of targeted sequences with increasing phylogenetic divergence. Furthermore, high substitution rates (saturation) and among-lineage heterogeneities in base composition may yield phylogenetic artefacts[31,36].

In this study, to alleviate potential ambiguities in site homology and account for potential sources of phylogenetic bias, we targeted only protein-coding regions to resolve the ordinal-level phylogeny of mosses. We designed oligonucleotide gene-baits to enrich genomic libraries for protein-coding genes from all genomic compartments (i.e., nuclear, plastid, and mitochondrial) across the Bryophyta targeting a broad taxon sampling to weaken effects of saturation on phylogenetic inferences[37,38], to address five critical areas of the moss phylogeny that have been previously contentious: (1) the earliest splits giving rise to the Takakiophytina, Sphagnophytina, Andreaeophytina, Andreaeobryophytina and Bryophytina; (2) the relationships of the nematodontous Tetraphidopsida and Polytrichopsida; (3) the early divergences among arthrodontous mosses; (4) the identity of the sister-group to the most speciose lineage, the pleurocarps or Hypnanae; and (5) the delineation of major lineages within the Hypnanae, which addresses the inconsistent placement of Hypopterygiaceae within the pleurocarpous mosses[25,39]. Our phylogenetic inferences based on 105 nuclear and 122 organellar protein-coding genes sampled for 142 species from 29 of the 30 moss orders highlight the evolutionary significance of taxa with reduced morphologies, shed light on the relative tempo and mechanisms underlying major cladogenic events, and resolve the relationships and delineation of moss orders. Our comparison of relative rates of substitutions across genomes further reveal that mosses exhibit the highest relative rate in non-synonymous substitutions in the nuclear loci, among divisions of land plants.

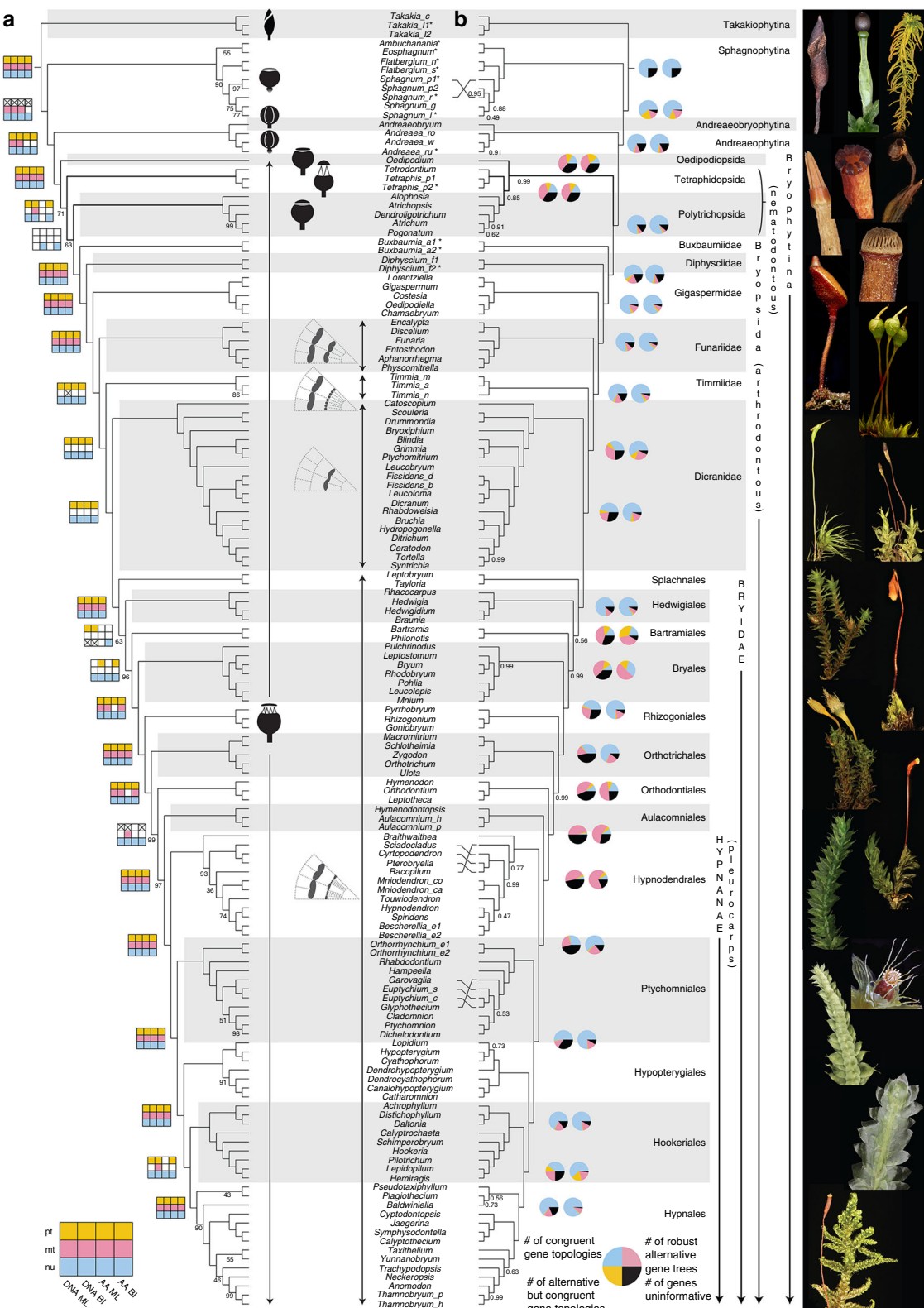

## Results and Discussion

**Deep splits using targeted sequences from all compartments.** The reconstruction of higher-level phylogenies using hundreds to thousands of protein-coding sequences has led to a re-evaluation, and deeper understanding of many fundamental relationships across the diversity of life. For example, transcriptome-based

phylogenomic analyses have contributed to resolving recalcitrant relationships in diverse groups such as molluscs[40], diatoms[41], embryophytes[42], and insects[43]. One advantage to phylogenomic approaches is the ability to identify and quantify conflicting signal for specific nodes among individual gene trees. Like several relationships that we discuss here (e.g., the branching order of

**Fig. 1** The ordinal relationships of moss tree of life (liverwort outgroup pruned for clarity). **a** Phylogenetic relationships from 105 concatenated nuclear single-loci amino acid sequences based on RAxML analyses (Supplementary Fig. 20); all branches maximally supported (i.e., 100% bootstrap frequencies) unless otherwise marked. Supports from inferences of the plastid (pt), mitochondrial (mt) and nuclear (nu) DNA, and amino acid (AA) sequences by maximum likelihood (ML), and Bayesian inferences (BI) were marked on the main nodes, a square with filled color indicates a strong support on the node (ML-BS ≥ 95; BI-PP ≥ 0.99); a square with a cross indicates a conflict with strong support; an empty square indicates conflict lacking strong support; **b** Coalescence based inferences of these nuclear single-loci amino acid sequences; ASTRAL tree with local posterior probabilities (Supplementary Fig. 24), branch lengths in coalescent units (2 × N generations) and are directly proportional to the amount of discordance; PhyParts Pie Charts of DNA (left) and AA (right) gene trees. The central sporophyte outlines characterize the major dehiscence types and their phylogenetic distribution; major arthrodontous peristome architectures are illustrated by sections of 1/8th of the amphithecium (see ref. [6]). Asterisk indicates exemplars for which transcriptome data complemented data recovered via targeted enrichment (see method section for details). Image credit for *Sphagnum*: Dr. Adam Wilson (University at Buffalo). All other images in Fig. 1 were taken by the senior author (B. Goffinet)

Takakiophytina and Sphagnophytina), discordance among gene trees due to processes such as incomplete lineage sorting, ancient hybridization, or differential loss of paralogs helps explain why these relationships have been difficult to resolve. Challenges to resolving the relationships of columnar cacti, for example, are consistent with extensive disagreement between gene histories and the species history[44]. The advent of target-enrichment methods, such as anchored hybrid enrichment[45] or more generalized methods (e.g., Mandel et al.[46]), has increased the efficiency and cost-effectiveness with which phylogenomic datasets can be gathered. Conflict among relationships inferred from nuclear genes and those inferred from organellar genes may also be illuminating (for a recent example, see Morales-Briones et al.[47]). Organellar datasets may be recovered passively from off-target reads[48] or actively, using probes specifically designed to enrich libraries for organellar regions. Few plant phylogenetic studies evaluate conflicting topologies among all three genomes (nuclear, plastid, mitochondrion) and it is uncommon to actively capture these datasets concurrently using target enrichment (e.g., Bogarín et al.[49]). To our knowledge, our study represents the only case in which complete plastid and mitochondrial gene sets have been analyzed alongside a large set of nuclear genes to resolve higher-order relationships.

**Efficiency of target enrichment**. We targeted 40 mitochondrial, 82 plastid, and 150 nuclear loci for 142 taxa (Supplementary Table 1). As previously shown[27,50], multiplexing 96 instead of 36 libraries in a single hybridization reaction did not result in a reduction in gene recovery rate (Supplementary Figs. 1–4). Even using a low-output Illumina MiSeq system, high gene recovery was achieved for both the organellar and the nuclear genes. Almost all plastid loci (average of 78.8 genes or 96.1% of loci with a recovery of any target length and 78.0 genes or 95.1% for loci recovered at 50% of target length, Supplemental Fig. 1) and mitochondrial loci (average of 39.9 genes or 99.8% of loci recovered at any target length and 39.4 genes or 98.5% for loci recovered at 50% of target length, Supplemental Fig. 2) were successfully captured across all samples. Fewer loci were captured on average from the nuclear genome (127.2 genes or 84.8% for loci recovered at 50% of the target length and 135.9 genes or 90.6% for any level of recovery, Supplemental Fig. 3), although the recovery rate for arthrodontous mosses was higher, averaging 134.5 genes or 89.7% at 50% percent recovery or 141.3 genes or 94.2% if any portion of the gene is recovered. The capture rate for nuclear loci is comparable to those obtained via anchored enrichment methods targeting conserved nuclear regions (88.4–99.8%; 88.6%)[27,45], or nuclear protein-coding genes (90.3%)[50].

While the recovery rate appeared almost constant across the moss phylogeny (Supplementary Fig. 5), it decreased substantially for nuclear loci with increasing phylogenetic distance between the sampled taxon and the reference taxa used for designing baits,

e.g., 150 (*Trachypodopsis*) to 16 (*Takakia lepidozioides* 2) (Supplementary Fig. 5). Baits for organellar loci were designed from orthologous sequences from across bryophytes and even vascular plants (Supplementary Table 2), whereas those for nuclear loci were based on inferred single-copy genes in the *Physcomitrella patens* genome and at least one homologous sequence from a pleurocarpous moss transcriptome generated by the 1KP project[51]. Coupled with the higher rates of evolution of nuclear vs. organellar genes[29,30], this may explain the lower rate for gene capturing and recovery for nuclear loci. For example, we identified 73 homologs of our targeted nuclear loci in the 1KP project transcriptome data in *Sphagnum*, with an average pairwise p-distance (PWD) to *P. patens* of 0.32; in contrast, we only recovered 20 loci with an average PWD of 0.25 with HybSeq (Supplementary Fig. 6). The addition of transcriptome data for *Ambuchanania*, *Eosphagnum*, and *Flatbergium* considerably reduced the missing data for Sphagnopsida; at least two samples of this clade were present in 99 of 105 gene trees. Similarly, for *Takakia*, we recovered 70 loci from 1KP homologs, but only 21 loci with HybSeq (Supplementary Fig. 6). The addition of transcriptome data allowed for at least one sequence from *Takakia* in 81 of 105 gene trees. Across the non-peristomate mosses for which we had both 1KP and HybSeq data, the recovery efficiency via HybSeq dropped dramatically when the PWD to *P. patens* was above 30% (Supplementary Fig. 6). Similarly, the virtual lack of enrichment for liverworts necessitated the inclusion of nuclear sequences recovered from the 1KP project transcriptomes for phylogenetic reconstruction. Thus, to enhance the recovery rate across the phylogenetic breadth of mosses and outgroups, available 1KP transcriptome or other genomic data for these taxa should in the future be integrated when designing or optimizing baits. However, missing data, which tend to be common in phylogenomic datasets, may not have significant effects on phylogenetic inferences[52,53].

The nuclear probe set initially targeted 150 loci, but only 105 were used in the final analysis. The remaining genes were removed due to gene duplications within the Dicranidae or other abnormally long internal gene tree branch lengths suggestive of unclear orthology. Following realignment and the removal of misaligned portions, the mitochondrial data yield a concatenated alignment of 31,665 base pairs (bp) with 36.9% parsimony informative characters (PICs), the plastid 64,368 bp with 51.1% PICs, and nuclear 154,548 bp with 68.9% PICs.

**Evolutionary rates**. Within mosses, we recovered almost complete exomes from the plastid and mitochondrial genome, along with 105 nuclear genes for an overlapping set of mosses, allowing for the comparison of substitution rates across the three genomic compartments. Based on the synonymous (dS) and non-synonymous (dN) substitution rates for each gene, measured as the total tree depth in units of dS and dN, the relative ratio across the genomes (mitochondrial:plastid:nuclear; hereafter mt:pt:nu) is

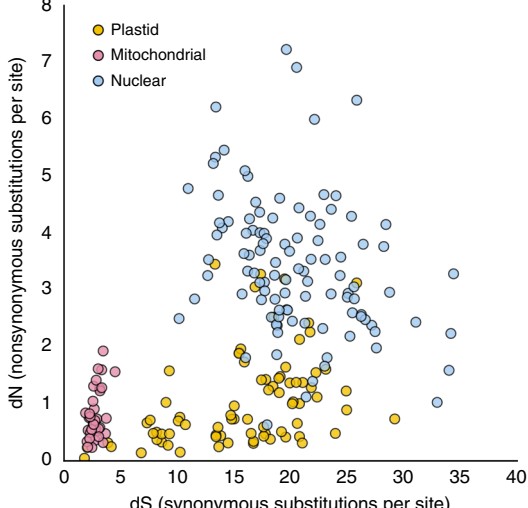

**Fig. 2** Total gene tree depth in synonymous (dS) and non-synonymous (dN) substitutions per site for protein-coding genes in three genomic compartments across Bryophyta. Units are synonymous (dS) or non-synonymous (dN) substitutions per site. Rates were calculated using individual gene trees in PAML with the liverwort outgroups removed. Rates for individual genes can be accessed in the online version of the figure (https://plot.ly/~mossmatters/15/)

1:5.7:7.4 and 1:1.3:4.2, respectively. Thus, mitochondrial genes in mosses evolve at the slowest rate, which is consistent with patterns observed for other plant lineages (Supplementary Table 3)[29,30]. Substitution rates in nuclear and plastid genes are typically higher, but also more variable (Fig. 2), which is consistent with observations in seed plants[30]. The lower substitution rate of organellar genes is likely a reflection of the higher mutation restoration efficiency of the organellar genomes, or stronger functional constraints on the few genes still remaining in the organellar genomes when compared to the gene complement of the endosymbiotic ancestors[54]. Indeed, genes from plastid and nuclear genomes accumulate synonymous substitutions at more similar rates compared to the mitochondrial genome (i.e., 1:5.7:7.4) but plastid genes generally exhibit rates of non-synonymous substitutions much lower than the nuclear genes, and at a rate more similar to mitochondrial genes (1:1.3:4.2; Supplementary Table 3 and Fig. 2). Some plastid genes (e.g., *ndh* genes involved in photorespiration, *cem*A and *mat*K) evolve at a rate similar to that more characteristic of nuclear genes (online version of Fig. 2; see legend). Only three genes (*psb*I, *psb*L, and *pet*L) evolved at mitochondrial-like rates for synonymous and non-synonymous substitutions. Many plastid genes, especially photosystem genes, including *psb*A, *psb*E, and *psb*F, have low rates of non-synonymous substitutions but have much higher rates of synonymous substitution compared to mitochondrial genes. The putative single-copy nuclear protein-coding genes exhibit the highest dN and dS rates, with over half of the 105 genes exhibiting higher dN than the 'fastest' evolving plastid genes (Fig. 2).

Our estimates of dS and dN are based on the phylogenetic tree rather than on pairwise comparisons, and hence refer to units distinct from, and not directly comparable to, those of previous studies by e.g., Palmer[29] and Drouin et al.[30]. Furthermore, our assortment of target capture nuclear genes is larger, non-overlapping with earlier studies, and perhaps under distinct selection regimes. When the data are normalized against the mitochondrial genome, the trend in the relative ratios in synonymous substitutions observed here for mosses is similar to that reported for seed plants (i.e., mt < pt < nu)[30], whereas the trend in the ratio in non-synonymous substitutions deviates markedly (mt < pt < nu vs. mt < pt ≈ nu; Supplementary Table 3). Relative to mitochondrial genes, plastid genes of mosses accumulate synonymous substitutions at a faster rate than seed plants (1:5.7 vs. 1:2.7), but non-synonymous mutations at a rate similar to that of gymnosperms and half that of angiosperms (1.3, 1.3, and 2.5, respectively). Synonymous substitutions in nuclear genes compared to mitochondrial genes accumulate slightly faster in mosses (7.4) than in seed plants (6.0), but slower than in angiosperms (10.0). By contrast, the ratio of the nuclear to plastid dS rate in mosses is slightly lower than that of gymnosperms (Supplementary Table 3). Mosses, however, exhibit the highest relative rate in non-synonymous substitutions in their nuclear genes (4.2), compared to all seed plants (1.7), with a rate relative to plastid genes (i.e., 3.2) much higher than that of angiosperms (i.e., 1.4) or seed plants in general (i.e., 0.9). Furthermore, the dN/dS ratio (ω) for nuclear genes is more than three times higher in mosses than in seed plants, an observation in conflict with an absence of significant differences reported previously[55] based on single genes from each genome. Relative nuclear substitution rates in mosses are, thus, higher than previously estimated and a priori incongruent with a concept of relative stasis[55] in the evolution of their nuclear genes. This hypothesis is consistent with the observation that selection is not more efficient in haploid organisms, such that at least the haploid specific genes are not subject to slower rates of evolution[56].

**Phylogenetic inference.** The ordinal phylogeny of mosses was inferred using Maximum likelihood and Bayesian methods on concatenated individual gene alignments for each genome (Fig. 1a), and using a summary coalescent method (ASTRAL-II) for nuclear gene trees (Fig. 1b)[57]. For each approach, the phylogenies using nucleotide sequences and the inferred amino acid sequences are described in Supplementary Table 4 and Supplementary Figs. 7–27. Below, we discuss discordance among trees inferred from different genomes (intergenomic phylogenetic conflict), among nuclear gene trees (intragenomic phylogenetic conflict), and among analyses for each of the five primary phylogenetic questions.

Our analyses of concatenated nuclear or mitochondrial loci strongly support Takakiophytina as emerging from the earliest split, followed by Sphagnophytina (Fig. 1 and Supplementary Figs. 11–14 and 18–21). This result is consistent with previous phylogenies based on 17 plastid genes and spacers[15] and a single-mitochondrial locus[58]. By contrast, inferences from the plastome resolve Takakiophytina and Sphagnophytina as a monophyletic group that is sister to all other mosses (Fig. 1 and Supplementary Figs. 7–10), a hypothesis consistent with previous inferences from variation in a few loci[13,14]. Bipartition analyses[59] of nuclear amino acid gene trees reveal 57 loci that are concordant with the *Takakia*-sister hypothesis, 10 with the *Sphagnum*-sister hypothesis, and 18 loci support the Takakiophytina-Sphagnophytina clade (Supplementary Fig. 28a). For nucleotide gene trees, 66 are concordant with the ASTRAL topology (i.e., *Takakia*–sister; Supplementary Fig. 28a). The most common alternative topology is the *Sphagnum*-sister hypothesis (11 genes) followed by the Takakiophytina-Sphagnophytina clade hypothesis (10 genes; Supplementary Fig. 28a). The *Takakia*-sister hypothesis is, thus, clearly favored by nuclear data, suggesting that the moderate support from morphological traits for the *Sphagnum*-sister hypothesis[12] is driven by character convergence.

Intergenomic conflict is sometimes interpreted as evidence of hybridization, which is phylogenetically widespread in mosses, including among extant species of *Sphagnum*[60]. However, intergenomic conflict as a result of hybridization would be at odds with the incongruence between signals from mitochondrial and plastid genomes, because it would suggest that the two organellar genomes were each inherited from distinct parents, rather than from the same parent as demonstrated for some mosses, including *Sphagnum*[61]. If hybridization did occur, it is not possible at present to determine whether the Takakiophytina or Sphagnophytina arose from such an event. Organellar leakage or relaxed organellar inheritance, which is known to occur in angiosperms[62], could also explain the incongruence between the plastid and nuclear genomes. Phylogenetic reconstruction from the organellar genomes may also be biased by shifts in RNA editing[63]. *Takakia* plastomes may hold extensive RNA editing sites[64] in contrast to the few that are found in *Physcomitrella*[65]. If a shift in the frequencies of such sites occurred early in the diversification of mosses (i.e., following the split of the Sphagnophytina), it may account for an artificial sister-group relationship between the Takakiophytina and Sphagnophytina.

Andreaeobryophytina and Andreaeophytina consistently compose a robust monophyletic group in all inferences (Fig. 1)[12,15,58] except from mitochondrial amino acids, which, however, provide only moderate support (Supplementary Figs. 13, 14). These lineages share a unique mode of dehiscence of their sporangium via four longitudinal slits, but differ in the origin of the tissue elevating the capsule, with the monospecific Andreaeobryophytina developing a sporophytic stalk rather than a gametophytic pseudopodium[66]. The position of the clade containing Andreaeobryophytina and Andreaeophytina relative to the Sphagnophytina and Bryophytina (both of which dehisce via the loss of an operculum) suggests that the dehiscence via four vertical lines arose once and potentially from an ancestral dehiscence mechanism through the loss of an operculum. However, the topology does not unambiguously resolve the evolution of the sporophytic seta, which is developed in *Takakia*, *Andreaeobryum* and all other mosses but not in the Sphagnophytina and Andreaeophytina, wherein the sporophyte is elevated on a pseudopodium[66]. Thus, neither state defines a unique monophyletic group, and the reconstruction of the ancestral state may depend on the assumption of homology to that of the liverwort seta, which follows a distinctive ontogeny[4]. However, the distinct ontogeny of the pseudopodium in the Andreaeophytina and Sphagnophytina, the occurrence of a pseudopodium in the highly derived *Neckeropsis*, and the development of analogous pseudopodia terminating in clusters of gemmae in *Aulacomnium* (both genera of the Bryophytina)[6] strongly suggest that pseudopodium development is homoplasious within mosses, and strengthen the hypothesis that the seta is ancestral in mosses.

The typically peristomate mosses, or Bryophytina, form a robust lineage sister to the combined Andreaeobryophytina-Andreaeophytina clade (Fig. 1). The monophyly of the Bryophytina (Fig. 1) is strongly supported by all inferences from concatenated data (Supplementary Figs. 7–21) and coalescence analyses (Supplementary Figs. 22–25), and the concordance among the vast majority of individual nuclear loci (Supplementary Figs. 26 and 27) further strengthens this established hypothesis[16,21,58]. Within the Bryophytina, the inferred relationships among the main lineages are, however, novel.

The early divergences within the Bryophytina gave rise to four lineages, Oedipodiopsida, Polytrichopsida, Tetraphidopsida, and Bryopsida (Fig. 1), whose relationships were typically poorly supported[16,21,36,58]. Inferences from concatenated plastid, mitochondrial, or nuclear loci suggest that the Oedipodiopsida, Polytrichopsida, and Tetraphidopsida (hereafter OPT), make-up

a grade subtending the Bryopsida (Fig. 1a and Supplementary Figs. 7–10 and 18–21). Support for these successive splits is weak when inferences are made from amino acid sequences, suggesting that the strength of the phylogenetic signal recovered from nucleotide sequences is likely shaped by substitution saturation in synonymous positions in deeply diverging lineages[31]. The average nuclear locus carries in fact little signal, but the most common signal (shared by fewer than 50% of the loci) supports the monophyly of the OPT, with the Tetraphidopsida and Polytrichopsida as sister lineages (Fig. 1b). These observations suggest that the divergences of the OPT may have been rapid, resulting in signatures of these events distributed among few nuclear loci.

The relationships of the OPT taxa relative to the Bryopsida are critical to assessing homology of the nematodontous peristome types developed in the Tetraphidopsida and Polytrichopsida and their significance in the evolution of the characteristic arthrodontous peristome. A plurality of nuclear amino acid gene trees resolve a monophyletic vs. a paraphyletic OPT (16 vs. 11 amino acid gene trees), whereas 17 nucleotide gene trees recover a paraphyletic OPT against 11 with a monophyletic OPT (Supplementary Fig. 28b). Such topological incongruence may reflect the effects of substitutional saturation, among-lineage GC content heterogeneity or codon usage bias on inferring deep-phylogenetic relationships from nucleotide vs. amino acid data, which may therefore be preferred for resolving ancient cladogenic events[31,36]. A plurality of amino acid (i.e., 15) and nucleotide (i.e., 11) gene trees resolve Tetraphidopsida and Polytrichopsida as sister groups (Supplementary Fig. 28b). Only eight amino acid and DNA gene trees resolve Polytrichopsida and Bryopsida sharing a unique ancestry (Supplementary Fig. 28b).

The monophyly of a clade combining the Tetraphidopsida and Polytrichopsida (Supplementary Figs. 1 and 28b) may suggest that their nematodontous peristomes, comprising either four massive or typically 32 short teeth, respectively, are homologous, a hypothesis first rejected[67] and then resurrected[58]. The position of the aperistomate *Alophosia*[68] as sister to the rest of Polytrichopsida, however, suggests that the nematodontous peristome in the Tetraphidopsida and Polytrichopsida may have independent origins, especially as the next dichotomy within the Polytrichopsida also includes an aperistomate lineage[19,68]. On the other hand, secondary peristome loss is common throughout the diversification of mosses[6] and a unique origin of the nematodontous peristome should not be rejected until further comparative ontogenetic and transcriptomic studies elucidate the genetic networks underlying the development of the nemato- and arthrodontous peristomes.

The nematodontous and arthrodontous peristome differ in their architecture but may share a fundamental sequence of cell divisions within the layers contributing to the peristomes[6]. In a small plurality of amino acid trees, the Tetraphidopsida and Polytrichopsida make up the sister lineage to the Oedipodiopsida (Fig. 1b) rather than the Bryopsida (16 vs. 11 amino acid trees, respectively; Supplementary Fig. 28b). The sole species of the Oedipodiopsida, long considered a member of the Bryopsida (or Bryales sensu[17]), exhibits unique traits of the sporophyte[69] and leaf ontogeny[70], and lacks a peristome[17]. Prior phylogenetic inferences resolved it either as sister to the Polytrichopsida[20] or to all peristomate mosses[12,13,15,67]. Ligrone and Duckett[71] argued for a unique shared ancestry for *Oedipodium* and all peristomate mosses to the exclusion of the Polytrichopsida, on the basis of traits of the placenta and water-conducting cells. None of our inferences resolve the Polytrichopsida as marking the earliest split, and hence their traits must reflect either strict apomorphies or result from reversals to states present in the earliest diverging mosses.

Within a scenario of a monophyletic OPT sister to the Bryopsida, the transformational relationships between the nematodontous and arthrodontous peristomes remain ambiguous due to the uncertainty of homology of the two main nematodontous types and of the ancestral state in the OPT, given that the Oedipodiopsida lack a peristome. Unless the peristome originated once in mosses and was secondarily lost in *Oedipodium*, the two main peristomial architectures of mosses would have independent origins and hence not be homologous.

All three genomic datasets strongly support the monophyly of the Bryopsida (Fig. 1), which typically share an arthrodontous peristome[17]. This hypothesis is consistent with prior inferences from few discrete loci[13,21]. The architecture and ontogeny of the teeth diagnose major lineages within the Bryopsida[17], but their characters alone fail to resolve the relationships among these lineages[6].

Our inferences confirm[16,21] that within the Bryopsida, the earliest splits gave rise to the Buxbaumiidae, and then the Diphysciidae (Fig. 1 and Supplementary Figs. 7–25). The next split involves the Gigaspermaceae, a family lacking a peristome and traditionally placed, based on vegetative similarities, close to the Funariaceae[17]. The first evidence against a uniquely shared ancestry between the Gigaspermaceae and the remainder of the Funariales consisted in a 71 kb inversion in the plastid genome of the Funariaceae, Disceliaceae, and Encalyptales but lacking in the Gigaspermaceae, which were, therefore, accommodated in their own order, Gigaspermales[72]. Their relationships remained, however, ambiguous[21]. Our inferences from concatenated plastid or nuclear loci (Supplementary Figs. 7–10 and 18–21), as well as coalescence analysis of nuclear loci (Supplementary Figs. 22–25) resolve the Gigaspermales as diverging after the Diphysciidae, and hence as sister to the remaining Bryopsida (Fig. 1). Only inferences from the mitochondrial genome challenge this result, resolving the Gigaspermales within the Funariidae, albeit with weak support (Supplementary Figs. 11–14). Our phylogenetic hypothesis (Fig. 1) is consistent with the recognition of the Gigaspermidae[18] and highlights again the phylogenetic significance of lineages with reduced morphologies during the diversification of mosses. These splits are statistically robust based on plastid and nuclear loci (and concordant among a majority of the latter), but are poorly supported based on mitochondrial loci, suggesting that the successive divergences occurred in a radiation too rapid for fixation of mitochondrial alleles, resulting in incongruence.

The splits following the divergence of the Gigaspermidae mark the origin of the Funariidae, Timmiidae, Dicranidae, and Bryidae, four lineages diagnosed by distinct arthrodontous peristome ontogenies and architectures (Fig. 1)[6]. The Dicranidae include *Catoscopium*, a genus historically treated as a member of the Bryidae[6], but resolved here as sister to the remainder of the subclass, corroborating the hypothesis originally proposed by Cox et al.[21] and then Ignatov et al.[70]. The relationships among Funariidae, Timmiidae, Dicranidae, and Bryidae suggest that alternate alignments of the inner and outer peristomial appendages, as in the Bryidae[17], are derived from a plesiomorphic opposite arrangement, as in the Funariidae and Dicranidae[73]. The Dicranidae are characterized by a single ring of teeth, the endostome, and diagnosed by an asymmetric cell division occurring early in the ontogeny of the inner peristomial layer[6]. Similar divisions occur in other lineages, such as in the earlier-diverging Diphysciidae[6] or the derived subclass Bryidae. Consequently, the phylogenetic significance of the asymmetric cell division is ambiguous. However, the consecutive divergence of the Funariidae and Timmiidae (first proposed by Chang et al.[16]), both with strictly symmetric divisions[6], suggests that the asymmetric division in the inner peristomial layers of the Dicranidae and Bryidae[6] is evolutionarily derived, as proposed by Vitt[17].

The monophyly of the Bryidae is robustly recovered by all genomic compartments and the vast majority of nuclear loci (74 of 96 amino acid gene trees; Fig. 1). This clade is diagnosed by its peristome ontogeny[6] and architecture[17]. Prior reconstructions of the phylogeny of the Bryidae relied on few discrete loci and often an incomplete sampling of the familial diversity, and yielded poorly supported[21] and conflicting hypotheses[21,22,58]. Despite our extensive locus sampling, the earliest splits, yielding the Splachnales, Hedwigiales, and Bartramiales, and ultimately the remainder of the Bryidae, are incongruent among inferences, and in some cases, poorly supported by concatenated data or not concordant among nuclear loci (Fig. 1). These topological ambiguities and weaknesses may be due to the rapid succession of the early divergences within the Bryidae. Enhancing taxon sampling and extending character sampling to more exons and possibly their introns may be needed to finally resolve these relationships. The remainder of this subclass composes a highly robust clade, consistent among a plurality of gene trees, with the first splits segregating the Bryales and then the Rhizogoniales (Fig. 1).

The diversification of the Bryidae led to the origin of the superorder Hypnanae, or core pleurocarps, a robust lineage (Fig. 1) characterized by the development of female sex organs on short lateral branches[5]. The Orthotrichales, Orthodontiales, and Aulacomniales complete the grade of largely acrocarpous Bryidae subtending the Hypnanae, whose monophyly is supported by concatenated nuclear data and a plurality of loci (Fig. 1). Phylogenetic signal from the mitochondrial loci is mostly weak or congruent with this grade, but plastid loci support an alternative hypothesis (Fig. 1), whereby the Orthotrichales and Orthodontiales share a unique ancestor, sister to the Aulacomniales and core pleurocarps (Fig. 1). Such incongruence between plastid vs. mitochondrial and nuclear data is similar to that observed in the case of *Takakia* and *Sphagnum* and could also be explained by ancient hybridization.

The hypothesis that the Aulacomniales are the sister-group to the Hypnanae[22,74] is highly supported and consistent among all analyses of concatenated data. Although only two amino acid and eight nucleotide nuclear gene trees provide signatures of such shared ancestry, no alternative signal is found in more than three gene trees among nuclear loci. Such a pattern is expected under the multispecies coalescent model—when coalescent branch lengths are short (including rapid divergences), the most common gene tree topology may disagree with the species tree topology[75]. Species tree methods like ASTRAL have been shown to be consistent under the multispecies coalescent, resulting in a combined analysis with accurate reconstruction even with high incongruence among individual gene trees. This illustrates the utility of genome wide character sampling. In fact, resolving the relationships based on few discrete loci would harbor limited informative substitutions, as evident from the very short branch subtending the combined Aulacomniales and Hypnanae clade in all analyses (Supplementary Figs. 7–21).

The Hypnanae currently comprise four orders[6], and inferences have converged to the Hypnodendrales being sister to the remaining pleurocarps[74], accommodated in the Ptychomniales, Hookeriales and Hypnales[21]. The Ptychomniales were traditionally included in the Hypnales[17] until Buck et al.[25] resolved them as the sister-group to the Hookeriales and Hypnales. Our phylogenomic inferences unambiguously broaden the circumscription of the Ptychomniales to include two taxa that were always included in the Hypnales, the Orthorrhynchiaceae and the monospecific genus *Rhabdodontium* of the Pterobryaceae sensu Goffinet et al.[6], with both taxa consistently emerging from the

earliest splits within the Ptychomniales (Fig. 1). Members of both taxa are characterized by ecostate leaves, in contrast to the synapomorphic double costa of the Ptychomniales or its sole family, the Ptychomniaceae[25].

The Orthorrhynchiaceae should be maintained as distinct within the Ptychomniales. The aquatic *Rhabdodontium buftonii*, which is the sole species of the genus, has been classified in the Pterobryaceae (Hypnales) since its establishment. It is consistently shown here to be the sister taxon to the Ptychomniaceae, and is on the basis of this relationship and its lack of the synapomorphies defining the Ptychomniaceae, accommodated in its own family: Rhabdodontiaceae B. Mishler, N. E. Bell & P. J. Dalton fam. nov. (Plants pleurocarpous, stoloniferous, leafy axes julaceous, leaves ecostate partially multistratose, capsules immersed, exostome striate below, and striate exostome teeth; Type: *Rhabdodontium* Broth., *Nat. Pflanzenfam.* 1(3): 803 (1906)).

The Hypopterygiaceae are also resolved as a sister-group to the Hookeriales and Hypnales, rather than sharing a unique ancestor with only the Hookeriales[25,39]. Our hypothesis is strongly supported by concatenated plastid and nuclear data and amino acid sequences of most nuclear genes (Fig. 1). To reflect this topology, wherein the Hypopterygiaceae make up a lineage distinct from the other orders, we propose to raise them to the ordinal rank: Hypopterygiales Goffinet ord. nov. (Plants pleurocarpous, with shoots differentiated in stolons and stems, stems distally heterophyllous and with 1/3 or nearly so phyllotaxy; Type: Hypopterygiaceae Mitt., *J. Proc. Linn. Soc., Bot.*, Suppl. 2: 147 (1859)).

### Targeted enrichment of coding sequence.
The enrichment of genomic sequencing libraries for targeted regions using RNA probes or baits is an efficient and cost-effective way to generate hundreds to thousands of phylogenetically informative sequences with relatively dense taxon sampling[76]. While this approach is becoming more common as a source of data for phylogenetic inference, it is still uncommon to apply this method to resolve higher-order relationships of diverse groups with relatively ancient origins, such as mosses. Target-enrichment approaches for diverse groups, such as butterflies[77] highlight the potential for resolving historically difficult relationships or, at the very least shedding light on why resolving some relationships is a recalcitrant problem. We applied a target enrichment approach to sequence nearly complete plastid and mitochondrial gene sets alongside 105 nuclear genes for 134 species. In addition to allowing for the comparison of relative rates of substitutions across genomes, our phylogenomic analyses show that this method increases confidence in challenging areas of the moss tree of life, providing robust hypotheses for most portions of the ordinal moss tree, while demonstrating that intra- and intergenomic signal conflict underlays relationships that have been, and may continue to be, difficult to resolve.

## Methods
### Taxon sampling.
To reconstruct the ordinal phylogeny of mosses, we sampled 134 species from 124 genera, 64 families and all but one of the 30 orders spanning the moss tree of life[6] (Supplementary Table 1). Three species of liverworts were chosen as outgroups to root the tree. For these outgroups, the organellar genes were gained from a NGS massively parallel sequencing data pool (*Bazzania* and *Scapania*) or the published organellar genomes (*Marchantia*) from the NCBI's genome database (http://www.ncbi.nlm.nih.gov/genome), and nuclear genes were extracted from the 1000 Plant Transcriptome Project (1KP Project, http://www.onekp.com)[51].

### Gene selection and bait design.
We targeted 272 protein-coding genes, including the set of 122 organellar genes (82 plastid and 40 mitochondrial) commonly shared by land plants, and 150 single-copy nuclear genes predicted by comparing available moss genomes and transcriptomes. The enrichment of genomic libraries requires stoichiometric precision in that all genomic sequences that are homologous to an

RNA bait are competing for that 'bait space' during the hybridization phase. Therefore, genes that are represented more frequently in the genomic library, high-copy organellar DNA for example, may hybridize more efficiently than low-copy nuclear sequences. So, we created two separate sets of baits: the organellar and nuclear ones. In each set of baits, multiple taxa were used to design redundant baits with the objective of enriching libraries from the most phylogenetically diverse set of samples. Multiple genomic libraries, therefore, could be hybridized to the same set of baits and multiplexed in a single reaction. Pooled libraries were enriched once in organellar genes (mitochondrial and plastid together), and once in the selected nuclear genes. Organellar baits were designed based on reference sequences of plastid and mitochondrial genomes of multiple mosses, and other land plants, including liverworts, hornworts, lycophytes, ferns, gymnosperms, and angiosperms (Supplementary Table 2). For nuclear baits, single or low-copy nuclear genes were selected based on the genome sequence of *Physcomitrella patens* (Funariales; Supplementary Table 2) and transcriptomes of five Hypnales (*Anomodon attenuatus, A. rostratus; Thuidium delicatulum, Climacium dendroides, and Hypnum subimponens*) available from the 1KP Project[51] (Supplementary Table 2), and matching baits were designed based on the selected reference sequences. In *Physcomitrella*, the targeted organellar genes comprise ca. 102 kbp, and nuclear genes add up to 224 kbp. Both the organellar and nuclear baits were 120 bp long and designed with a 2× tiling density (tiled every 60 bp). The bait sequences are available from the Dryad Digital Repository: https://doi.org/10.5061/dryad.tj3gd75.

### NGS sequencing.
Genomic DNA was extracted from ~0.4 to 4.0 g fresh or dried gametophytic or sporophytic moss tissue using the NucleoSpin Plant midi DNA extraction kit (Macherey-Nagel, Düren, Germany). For most moss samples, we used dried herbarium specimens with ages typically ranging from several months to 3 years old, and with a maximum of 15 years (i.e., *Andreaeobryum macrosporum*). DNA quantity was assessed using the Qubit fluorometer system (Invitrogen, Carlsbad, CA, USA). The genomic DNA samples (200 ng in 52.5 μL) were sonicated individually using the Covaris M220, or as a 96-sample plate using the Covaris E220 to a fragment size of ~550 bp. NGS libraries were prepared using the Illumina TruSeq Nano DNA Library Preparation kit (Illumina, San Diego, CA, USA). We tested multiplexing 36 and 96 libraries at equal molarity in a single hybridization reaction, and then enriched for the targeted genes using the custom designed MYbaits kit (MYcroarray, Ann Arbor, MI, USA) following the manufacturer's protocol. After enrichment, the libraries have an insert size between 400–1000 bp and 600 bp on average. The enriched, pooled libraries were paired-end sequenced on an Illumina MiSeq platform using the 600-cycle v3 sequencing kit (Illumina, San Diego, CA, USA) in the Center for Genome Innovation at the University of Connecticut at Storrs.

### Data processing.
The demultiplexed raw reads were downloaded from the Illumina BaseSpace web server, and trimmed by Trimmomatic (www.github.com/timflutre/trimmomatic) using the following parameters: LEADING:10 TRAILING:10 SLIDINGWINDOW: 4:20 MINLEN: 36. Only paired reads were kept for the assembly. If a sample had been sequenced more than once, reads from distinct runs were merged.

The targeted genes were extracted using a customer designed pipeline HybPiper (https://github.com/mossmatters/HybPiper)[11]. The pipeline first maps the trimmed reads to the reference genes using BLASTX (www.blast.ncbi.nlm.nih.gov/Blast.cgi), which uses amino acid sequences as a reference, or BWA (www.github.com/lh3/bwa), which uses nucleotide sequences. Then the mapped reads for each gene are separately assembled into contigs using SPAdes (www.github.com/ablab/spades). The assembled contigs are aligned to the reference protein sequence using Exonerate (www.github.com/nathanweeks/exonerate). The HybPiper package includes a Python script (retrieve_sequences.py) to extract genes from the pipeline results and create multi-sequence FASTA files for each gene. It also includes scripts (get_seq_lengths.py and gene_recovery_heatmap.R) to create heatmaps to visualize the recovery efficiency for genes, and a Python script (depth_calculator.py) to summarize the sequence depth. The pipeline has several quality control functions, including the ability to detect and investigate potential paralogous or contaminant sequences. To filter out the potential low-level contaminations, the pipeline drops assembled contigs with average depth <10 reads.

HybPiper identifies a single contig corresponding to each reference sequence. However, if paralogs exist, SPAdes might assemble multiple contigs that each represents the entire target sequence. In such case, HybPiper uses a two-step strategy to choose among multiple full-length contigs: first a sequencing depth cutoff—if one contig has a sequencing depth ten times (by default) greater than the next best full-length contig, it is chosen—and then a similarity criterion—if the sequencing depth is similar among all full-length contigs, the percent identity with the reference sequence is used. HybPiper will generate warnings indicating that multiple long-length matches (at least 75% of the target length) to the reference sequence have been found. Using HybPiper, all putative paralogous copies were extracted for each gene, along with the sequence generated from the main HybPiper script. Using gene trees constructed from all sequences, putative paralogs can be sorted into two categories: Type I paralogs that are monophyletic within species (possibly recent duplicates or alleles), and Type II paralogs that are the result of ancient gene or genome duplications. As the target probes were designed

from two fairly divergent taxa, the genes may not be single-copy in all groups of mosses. Genes were retained with a random copy selected if all putative paralogs were Type I. If Type II paralogs were found in five or fewer taxa, the gene was retained with sequences from those taxa removed. The final nuclear gene dataset included 105 nuclear coding genes.

Each gene was aligned with a local version TranslatorX (www.translatorx.co.uk). The program first translates the nucleotide sequence into amino acids using the standard genetic code, then uses MAFFT (www.mafft.cbrc.jp/alignment/software) to create an amino acid alignment; ambiguous portions were trimmed from the alignment by GBLOCKS (www.molevol.cmima.csic.es/castresana/Gblocks.html) with the least stringent settings. The cleaned amino acid alignment is then used as a guide for nucleotide sequence alignment. About 16–26% data were trimmed from the original alignments, resulting in alignments of mitochondrial, plastid, and nuclear loci, containing 1.4%, 3.4%, and 21.2% of missing characters, respectively. Stop codons were then removed from the resulting data matrix with the Perl script ReplaceStopsWithGaps.pl (https://gist.github.com/josephhughes/1167776). The individual gene matrices were concatenated into a master data matrix, and converted into Phylip and Nexus formats in Geneious v7.1.5 (Biomatters, New Zealand). The final alignments are available from the Dryad Digital Repository: https://doi.org/10.5061/dryad.tj3gd75.

**Phylogenetic analyses of concatenated data matrix**. The concatenated nucleotide and the corresponding translated amino acid datasets were analyzed with maximum likelihood (ML) and Bayesian inference (BI) under a homogeneous model, i.e., assuming composition homogeneity among taxa. For the nucleotide dataset, ML analyses were performed using the parallel version of RAxML v7.2.3[78]. The ML trees were calculated under the GTR + Γ model. Non-parametric boot-strap analyses were implemented by GTR-CAT approximation for 300 pseudo-replicates. We used PartitionFinder v1.1.1 (www.robertlanfear.com/partitionfinder) to examine the optimal partitioning scheme of the data. The partitions were defined a priori on the basis of codon positions, and a three-partition strategy corresponding to the three codon positions was selected for each dataset based on the Bayesian Information Criterion (BIC). Bayesian inferences using a GTR + Γ substitution model were conducted using MrBayes v3.2[79] based on the same partitioning strategy as used in ML analysis. A Markov Chain Monte Carlo (MCMC) was run for 5 million generations in each analysis, and substitution model parameters were unlinked among the partitions, so that they were estimated independently for each partition. In all analyses, branch lengths and topology were linked. Burn-in and convergence were assessed using the likelihood of the samples plotted against generation time and by monitoring diagnostics within and between chains from the estimated posterior distribution. Posterior probabilities (PP) of clade support were estimated by sampling trees from the posterior distribution after removal of the burn-in samples. All estimates of marginal likelihoods were computed in P4, which implements equation 16 of Newton and Raftery[80]. For the amino acid dataset, the optimal partitioning scheme was examined using PartitionFinder with a priori partitions defined based on genes. The plant organellar amino acid specific amino acid substitution models, gcpREV[81] and stmtREV[31] were evaluated. Partitioning strategies of 16, 8, and 93 were chosen by BIC for the plastid, mitochondrial, and nuclear dataset, respectively. ML and BI analyses were carried out using RAxML with 300 bootstrap replicates, and MrBayes analyses with 5 million generations, under each selected partitioning strategy.

As among-site composition heterogeneity may cause phylogenetic artifacts[82], we performed MCMC analyses using the CAT + GTR + Γ model implemented in PhyloBayes MPI v1.2d[83] to test for among-site compositional heterogeneity in the amino acid data for each dataset.

**Coalescent analyses of nuclear genes**. We used ASTRAL-II[57], a quartet-based method that is consistent under the multispecies coalescent to estimate the species tree from nuclear gene trees. Each nuclear gene tree was generated using RAxML, including 200 bootstrap replicates. Amino acid trees were generated under the PROTCATWAG model, and nucleotide trees were generated using GTRCAT, with a separate partition for the third codon position. In order to reflect gene tree uncertainty in the ASTRAL-II analysis, gene tree topologies were collapsed when gene tree bootstrap support was below 33%[84]. Support on the ASTRAL-II phylogeny was assessed using multi-locus bootstrap (MLBS), which samples the gene tree bootstrap phylogenies 100 times, and by the more recently developed local posterior probability (LPP) method, which estimates relative quartet support on each branch.

Recent phylogenomic analyses highlighted that traditional metrics of support, such as bootstrap frequencies, may suggest high levels of support despite significant among-locus incongruence[59]. We assessed the level of nuclear gene tree conflict with a bipartition analysis using PhyParts (bitbucket.com/blackrim/phyparts)[59,85]. The method assesses, for each node on a species tree, how many genes are concordant, in conflict, or without information for that bipartition. The method also separates the conflicting gene trees to determine if there is one frequently occurring alternative bipartition. We used pie charts on each node of the species tree to summarize and visualize gene tree bipartition support using the ETE Python package (www.ete-toolkit.org), and custom scripts available at github.com/mossmatters.

**Relative rate analyses**. We used PAML (www.abacus.gene.ucl.ac.uk/software/paml.html) to estimate the rate of synonymous and non-synonymous substitutions in 105 nuclear, 82 plastid, and 40 mitochondrial protein-coding genes. Individual gene trees were estimated for each in-frame nucleotide alignment gene as described above using RAxML. The outgroup sequences (liverworts) were removed for the substitution rate analysis. The codeml program in PAML was used to optimize the branch lengths and calculate a single synonymous (dS) and non-synonymous rate (dN) across all Bryophyta (codeml model M0). The python package ETE was used as a wrapper to run PAML and extract the rates from each gene.

## Data availability

The sequences reported in this paper have been deposited in the NCBI Sequence Read Archive (SRA; accession no. SRP118564, SRP128062). Information about the target capture gene set, gene recovery statistics, multiple sequence alignments, phylogenetic trees are available from the Dryad Digital Repository: https://doi.org/10.5061/dryad.tj3gd75]. All other relevant data are available from the authors.

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

## Acknowledgements

This study was made possible through financial support from the US National Science Foundation (grants DEB-1240045 to BG; DEB-1239992 to N.J.W.; DEB-1239980 to A.J.S.), from the Fundação para a Ciência e a Tecnologia (FCT), Portugal (PTDC/BIA-EVF/1499/2014 to C.J.C.), as well as from the National Natural Science Foundation of China (grant 31470314 to Y.L.). DEB-1146168 to B.A. covered the specimen acquisition. For their contribution of specimens, we are also grateful to Hiroyuki Akiyama (Museum of Nature and Human Activities, Hyogo, Japan), William R. Buck (New York Botanical Garden), Patrick Dalton (University of Tasmania, Australia), Judith Harpel (University of British Columbia, Canada), Yu Jia (Institute of Botany, Chinese Academy of Sciences), Juan Larraín (Pontificia Universidad Católica de Valparaíso, Chile), Élisabeth Lavocat Bernard (Guadeloupe), Niklas Lönnell (University of Stockholm, Sweden), Jairo Patiño (Instituto de Productos Naturales y Agrobiología (IPNA-CSIC), La Laguna, Tenerife, Spain), Ann Rushing (Baylor University, TX), Steve Sillett (Humbold State University, USA), Li-Song Wang (Chinese Academy of Sciences, Kunming Botanical Garden, China), Patrick Williston (Smithers, British Columbia). We also wish to acknowledge the generosity of Mark Smith (Macroscopic Solution, CT) for his continuous assistance with the macropod photography, Dr. Adam Wilson (University at Buffalo) for the picture of *Sphagnum*, and of Dr. Yu-Kin He (Capital Normal University, Beijing, China) for providing fresh material of *Takakia* needed for the illustration. Dr. Wei Wang (Institute of Botany, Chinese Academy of Sciences) kindly commented on an earlier draft of the manuscript. Our probe design was made possible thanks to early access to transcriptome data from the One-Thousand Plant Transcriptome (1KP) Project, facilitated by Gane Wong, Jim Leebens-Mack, and Sean Graham. The Royal Botanic Garden Edinburgh (N.E.B.) is supported by the Scottish Government's Rural and Environment Science and Analytical Services Division, and we are also grateful for the support in 2018 of players of People's Postcode Lottery toward our scientific research.

## Author contributions

Y.L., M.G.J., A.J.S., N.J.W., and B.G. designed the project; B.G., N.J.W., & A.J.S. designed and carried out taxon sampling; Y.L., M.G.J., N.D., R.M. generated all the data; Y.L., M.G.J., & C.J.C. completed all analyses; J.R.S., L.H., A.V., N.E.B., B.A., & D.Q. provided samples for critical taxa; Y.L., M.G.J., C.J.C., A.V., A.J.S., N.J.W., & B.G. wrote the paper.

## Additional information

**Competing interests:** The authors declare no competing interests.

