## [Peer Review File · Nature Communications]

Reviewer #1 (Remarks to the Author):

Review of “Resolution of the backbone phylogeny of mosses using targeted exons from organellar and nuclear genomes” by Y. Liu et al.

This is an exciting study to read for those interested in overall moss phylogeny, or in applying hyb-seq methods to the overall study of plant phylogeny. Mosses are one of the major land-plant clades, so this is an important advance, and the study should attract a wide readership at Nature Communications. The results are mostly focused on the phylogenetic results, and don't go into endless detail for the fine-scale relationships across different analyses. This is a sensible compromise as most of these are visible in the trees for interested readers. There is also a very interesting comparison of synonymous and non-synonymous substitution rates, which show some interesting parallels and differences to other land-plant groups. I do have some relatively minor methodological/interpretation questions:

Minor points

- p. 6. Saturation can lead to phylogenetic artefacts, but well-sampled phylogenies, perhaps including this one, are known to be extremely resistant to the distorting effects of even extremely high substitution rates. Seminal work by Hillis and colleagues on this point should be cited here.
- p. 7. The recovery rates look awesome to me. I recommend you highlight in the Abstract (should be of interest to others using this method or your baits).
- p. 7. Which pairwise distance measure (model) was used? Or are these just p-distances?
- p. 8 and Fig. 2. Please highlight which sequences are from transcriptomes (e.g., use different text color in the figure).
- p. 9, 10. I think the non-bryophyte rates listed in parentheses here come from published studies, but I am not 100% sure (footnote to table S3?). Can this be made really clearer in the main text?
- p. 11. The sentence starting “Relative nuclear substitution rates in mosses... suggest thus at least a potential...” struck me as a bit waffly/motherlode-ish statement (“suggest” “at least” “a potential” is three qualifiers in a row). And can't we have strong selection with slow rates (I am not sure that this association necessarily holds).
- p. 12. “it is not possible at present to determine whether the Takakiophytina and Sphagnophytina arose from such an event” (i.e., hybridization). ILD could be discussed here too as an alternative source of intergenomic conflict. One reading of the final sentence on p. 19 is that you have decided that hybridization is to blame, so I would re-word this (it is just not clear). ILD could also usefully be raised elsewhere (e.g., first sentence of p. 15?).
- p. 13. Chang et al. (2014) discussed how the arrangement of Takakiophytina and Sphagnophytina to each other and other mosses was sensitive to model details – address that here or in general? It is likely that no existing models are adequate, and more refined partitioning schemes could improve fit. But does it matter for phylogenetic inference here? How would we know? The partition models used here do seem a bit crude, as only three are considered (for the three different codon positions

in DNA analyses; partition analyses that consider both genes and codons show that such schemes are often not suboptimal compared to more refined schemes). Of course there are endless model complications to consider, but I am not sure that composition bias (addressed here) is the only or the most concerning complication. The possibility at least could be raised that other model details may matter in some critical cases here, even if not tested here.

- p. 13-14. Some of the discussion of ancestral state evolution would benefit from simple diagrams to show the possibilities (e.g., homoplasy vs. homology for pseudopodia), even if not explicitly mapped on full trees. People in the field will get this, but for communication outside the immediate field it would be useful to have simple visuals showing different possibilities of homology or convergence, here and elsewhere (e.g., later for different peristome types) as an aid to general readers in understanding these important points.

- p. 14. "Strength of the signal... reflects substitution saturation". Aren't "signal" and "saturation" contradictory?

- p. 15. Earliest splits in a phylogeny lead to ALL descendants, strictly speaking, so "the earliest splits in Polytrichopsida" necessarily lead not only to aperiostomate lineages, but to peristomate ones too, ultimately.

- p. 15. Explain a bit better what would be needed to show or refute parallel vs. homologous origins of nematodontous peristomes ("comparative ontogenetic and transcriptomes" is a bit vague).

- p. 16. How about stating a "small" plurality of trees (16 vs. 11 does not seem very different to me).

- p. 16. One scenario is listed about peristome loss at the end of this paragraph. For balance, please spell out the alternative briefly ("otherwise...").

- p. 17. The "Buxbaumiidae... Diphysciidae..." consecutive branching has been confirmed in other studies (e.g., summary in Fig. 2 of Chang et al. 2014) – how about noting that? And the "Funariidae... Timmiidae..." consecutive divergences (particularly Timmiidae placement) were I think nailed down first by Chang et al. (2014) in terms of being well supported, so also mention that on p. 18?

- p. 17. The 71 kb inversion is just a synapomorphy of a subset of Funariales, and so the inversion lacking Gigaspermales could still be the sister group of that group. In other words, this is an agnostic character when it comes to the possible existence of a clade comprising the two (it doesn't confirm or deny the possibility).

- p. 19. It is disappointing that some major relationships (Splachnales etc) are still ambiguous here, but hyb-seq won't solve all problems. Perhaps discuss what might be needed?

- p. 19. I am a bit uncomfortable with phrases like "the grade... leading to the Hypnanae" which make it sound like the latter are a pinnacle of evolution (and by implication other modern descendants are not). The point is that grades don't "lead" anywhere – they are artificial constructs of our taxonomic systems (clades are natural, not artificial, in contrast). Please avoid such "progressivist" language.

- p. 28. Where only these two AA substitution models considered??

- p. 26. Did you do any "spot inspections" and adjustments of alignments, or just completely trust the automated aligners?

- Figure S4. Would you expect high error rates on the genes with lowest coverage here? Is it a problem here?

- Figure S4. Are any of the “low coverage” taxa for nuclear genes at critical points in the phylogeny? There is nice simulation work by Wiens and others suggesting that sparsely occupied alignments are not necessarily a problem for phylogenetic inferences (perhaps cite this literature in reference to this?)

Tables and figures

- Table S3. LWL = ?
- Figs. S1-S3. Cell sizes seem uneven vertically (some lines not visible?) and taxon labels seem a bit offset from rows.
- Fig. S3. X-axis labels with gene names. Provide guiding text in the table header on how to look up what these are.

General wording issues:

- use ‘plastid’ (the general organelle) not ‘chloroplast’ (one kind of plastid)
- The phrase ‘Clade X composes Y’ is used many times in the MS. This is an awkward construction in English. More usually it would be written “clade X is composed of...” or “clade C comprises...”
- There are other constructions that need careful attention (often I think too much is crammed in one sentence and clauses could be broken up), such as the first sentence of the Abstract (very odd), and the sentence starting “These transformations...” on p. 2. There are other examples that should be identified and tackled by a skilled English writer.
- Please use “amino acid” rather than “peptide” for AA alignments or substitution models (the former is much more commonly used, and is actually used in some of the Suppl. Info. (e.g., S24 legend))

Specific wording issues:

- p. 2. ‘maintaining sexual reproduction’ or ‘allowing’ it?
- p. 4. Avoid ‘since’ for non-temporal comparisons
- p. 5. ‘Persistent’ not ‘persisting’
- p. 10. Do you mean ‘non-overlapping genes with earlier studies’?
- p. 12. Define ‘organellar leakage’
- p. 14. “no conflicting signal” ... conflicts... versus what?
- p. 14. I am not sure what “unique ancestry” means (monophyly?)
- p. 15. Avoid archaic “whereby”
- p. 15. Should this be “The monophyly of a clade comprising Tetrastichidae... etc”
- p. 18. “ambiguous” (line 1) – does this mean “poorly supported”?

- p. 19. “A highly robust and concordant clade” do you mean the clade itself (one branch to consider, the one supporting the clade... so what’s it concordant to?) or structure within the clade (if so, spell this out clearly).
- p. 25. “and then a similarity criterion...” – Words seem to be missing here.
- p. 29. Remove left parenthesis and space before “highlighted” on first line?
- Ref. 59. Italics needed.
- p. 57. Why is “moss backbone” capitalized? Some phylogeneticists are uncomfortable with “backbone” (here and title) as the whole tree is a bifurcating system, not a caterpillar-spine (e.g., the “spine” shown “leading” to major clades – those we chose to focus on -- in the figures the “spine” is just one of many possible trackings of bifurcating branches along a tree, from root to tip).

Reviewer #2 (Remarks to the Author):

This by far the most robust moss phylogeny to date. This had been a major vexed question dating back to the last century . Though some issues still remain, this paper is likely to become the standard reference for some years to come. It contains very good major novel data sets and very good taxon sampling. The discussion is well balanced and extremely well argued.

All this is fine if you are well acquainted with moss diversity , as I am as the reviewer, but it’s very dense heavy going if you are not. To make this article more readable and of wider appeal I feel that the authors should consider broadening the discussion to include mention of the phylogenies of other groups (plant and animal) comparable mosses. As it stands the focus is perhaps too narrow for Nature Communications.

Minor points;-

Abstract. There may be a strict word limit here but it would be better if the key findings were included.

Introduction change wording;-

The vegetative body has undergone developmental shifts to optimize vegetative growth while maintaining sexual reproduction.

Teleology change shifts to optimize to shifts that

mosses arose from repeated burst of diversification - typo bursts

that may be triggered by extrinsic factors such as changes in global climate

-wrong tense --may have been triggered

Responses to referees' comments on ms NCOMMS-18-23931-T:

We thank the reviewers for their constructive criticism. We have addressed all comments. In addition to addressing specific comments by the reviewers, we also clarified the use of the word coverage (alluded to by one reviewer in the context of figures), clarified and updated some of the supplementary figures, and enhanced the main figure (i.e., Fig. 1) by illustrating some additional lineages. Below we detail our response to each point raised by the reviewers.

Reviewer 1":

- p. 6. Saturation can lead to phylogenetic artefacts, but well-sampled phylogenies, perhaps including this one, are known to be extremely resistant to the distorting effects of even extremely high substitution rates. Seminal work by Hillis and colleagues on this point should be cited here.

Response: We are citing two recent studies that specifically address the bias introduced by saturation in nucleotide substitution. We assume that the reviewer is referring to the following publications:

- Heath TA, Zwickl DJ, Kim J, Hillis DM. 2008. Taxon sampling affects inferences of macroevolutionary processes from phylogenetic trees. *Systematic Biology* 57: 160–166.
- Hedtke SM, Townsend TM, Hillis DM. 2006. Resolution of phylogenetic conflict in large data sets by increased taxon sampling. *Systematic Biology* 55: 522–529.

We have now integrated these references in the last paragraph of the introduction: "We designed oligonucleotide gene-baits to enrich genomic libraries for protein coding genes from all genomic compartments (i.e., nuclear, plastid, and mitochondrial) across the Bryophyta targeting a broad taxon sampling to weaken effects of saturation on phylogenetic inferences (**Heath et al. 2008; Hedtke et al. 2006**), to address five critical areas of the moss phylogeny that have been previously contentious".

- p. 7. The recovery rates look awesome to me. I recommend you highlight in the Abstract (should be of interest to others using this method or your baits).

Response: We did address this but given the word limit imposed on the abstract, we reworded one sentence from "We targeted these 227 loci for 142 species from 29 moss orders, assessed relative rates of substitution, contrasted phylogenetic signals of organellar and nuclear loci, and assessed gene-tree conflict." to "We recovered an average of at least 98.5%, 95.1% and 84.8% of the 227 mitochondrial, plastid and nuclear loci, respectively, for 142 species representing 29 of the 30 moss orders." The abstract now slightly exceeds the word limit (i.e. 162).

- p. 7. Which pairwise distance measure (model) was used? Or are these just p-distances?

Response: Yes, we used pairwise dissimilarity, p-distance. We feel this measure is most relevant in this case because the hybridization between probe and genomic library relies on similarity, not evolutionary distance.

To clarify this, we added “p-distance” to the sentence “For example, we identified 73 homologs of our targeted nuclear loci in the 1KP project transcriptome data in *Sphagnum*, with an average pairwise p-distance (PWD) to *P. patens* of 0.32; in contrast, we only recovered 20 loci with an average PWD of 0.25 with HybSeq (Supplementary Fig. 6)” on p. 7.

- p. 8 and Fig. 2. Please highlight which sequences are from transcriptomes (e.g., use different text color in the figure).

Response: we address this pertinent comment by labelling taxa for which transcriptome data were used with an * and explain the symbol in the legend to Figure 1.

- p. 9, 10. I think the non-bryophyte rates listed in parentheses here come from published studies, but I am not 100% sure (footnote to table S3?). Can this be made really clearer in the main text?

Response: We do refer in the text on p. 9 to Palmer²⁹ and Drouin et al.³⁰ (the two main studies on this topic) and to Drouin et al.³⁰ in Table S3 where we report the rates they published. It is not clear to us how else this would be clarified.

- p. 11. The sentence starting “Relative nuclear substitution rates in mosses... suggest thus at least a potential...” struck me as a bit waffly/motherlode-ish statement (“suggest” “at least” “a potential” is three qualifiers in a row). And can’t we have strong selection with slow rates (I am not sure that this association necessarily holds).

Response: we reworded the sentence to: “Relative nuclear substitution rates in mosses are thus higher than previously estimated and a priori incongruent with a concept of relative stasis⁴² in the evolution of their nuclear genes.”

- p. 12. “it is not possible at present to determine whether the Takakiophytina and Sphagnophytina arose from such an event” (i.e., hybridization). ILD could be discussed here too as an alternative source of intergenomic conflict. One reading of the final sentence on p. 19 is that you have decided that hybridization is to blame, so I would re-word this (it is just not clear). ILD could also usefully be raised elsewhere (e.g., first sentence of p. 15?).

Response: on p. 12 we wrote: “If hybridization did occur, it is not possible at present to determine whether the Takakiophytina or Sphagnophytina arose from such an event.”

We reworded the sentence on page 19 to: “Such incongruence between plastid vs mitochondrial and nuclear data is similar to that observed in the case of *Takakia* and *Sphagnum* and ~~may also be due to~~ **could also be explained by** ancient hybridization.”

p. 13. Chang et al. (2014) discussed how the arrangement of Takakiophytina and Sphagnophytina to each other and other mosses was sensitive to model details – address that

here or in general? It is likely that no existing models are adequate, and more refined partitioning schemes could improve fit. But does it matter for phylogenetic inference here? How would we know?

Response: We agree entirely that the resolution of the relationships of Takakiophytina and Sphagnophytina are very dependent on the model - but this is also generally true of all difficult to resolve nodes. It is possible that greater partitioning could be a component of the solution to resolving this node but is unlikely to be the only necessary model aspect that needs to be explored - rate and composition heterogeneity across the tree, or even covarion rates, may need to be accounted for. But, these are all general considerations when faced with any difficult to resolve relationship. The only way we would know would be to try these other methods and investigate model adequacy and their effects. We propose to leave the text as is, rather to engage in a discussion on this topic.

The partition models used here do seem a bit crude,

Response: although we agree that the model may be a bit crude considering that multitudes of other partitioning schemes have been proposed, we consider it an obvious starting point for partitioning protein-coding genes, and considering the computational burden that more complex models would impose on analyzing our extensive data set we request to retain the text as is.

as only three are considered (for the three different codon positions in DNA analyses; partition analyses that consider both genes and codons show that such schemes are often not suboptimal compared to more refined schemes). Of course, there are endless model complications to consider, but I am not sure that composition bias (addressed here) is the only or the most concerning complication. The possibility at least could be raised that other model details may matter in some critical cases here, even if not tested here.

Response: Although we agree that among-site composition bias is certainly not the only aspect of the model that needs to be addressed, it is important for these data as we demonstrate. Whether it is the most important (or concerning) process that needs to be modeled is certainly debatable, but we would argue that it does not negate its need for modeling in these data. We also agree that other model details may be just as important as those we identify, but the same argument is relevant to every phylogenetic analysis that uses a model of substitution - it is after all axiomatic that the results depend on the model that was used, it being a simplified mathematical expression of a biological process, and as such should not need repeating each time a phylogenetic analysis is presented. We regard speculation here as to which other processes and constraints of change might be important to be of only minor relevance to our results (given that we carefully explain the models and assumptions that we use), and request not to have to engage in a discussion on potential modelling alternatives.

- p. 13-14. Some of the discussion of ancestral state evolution would benefit from simple diagrams to show the possibilities (e.g., homoplasy vs. homology for pseudopodia), even if not

explicitly mapped on full trees. People in the field will get this, but for communication outside the immediate field it would be useful to have simple visuals showing different possibilities of homology or convergence, here and elsewhere (e.g., later for different peristome types) as an aid to general readers in understanding these important points.

Response: Mapping some of the distribution of pseudopodia on the tree in Fig. 1 would in our view not be very helpful, all taxa referred to in the text (i.e., Sphagnum, Andreaea, Aulacomnium and Neckera) are already included in the tree and that further annotating the tree may not enhance its overall clarity.

As for the peristome, the clades of nematodontous and arthrodontous taxa are labelled as such, and the main types of arthrodontous peristome types are illustrated and used to characterize major clades. We have enlarged the diagram of peristome types in the figure to clarify the diagnostic value of these types.

- p. 14. "Strength of the signal... reflects substitution saturation". Aren't "signal" and "saturation" contradictory?

Response: we reworded the sentence to "suggesting that the strength of the phylogenetic signal recovered from nucleotide sequences is likely shaped by substitution saturation in synonymous positions in deeply diverging lineages³¹".

- p. 15. Earliest splits in a phylogeny lead to ALL descendants, strictly speaking, so "the earliest splits in Polytrichopsida" necessarily lead not only to aperistomate lineages, but to peristomate ones too, ultimately.

Response: We reworded the sentence to: "The position of the aperistomate *Alophosia*⁵⁵ as sister to the rest of Polytrichopsida, however, suggests that peristome teeth comprised of whole cells in the Tetraphidopsida and Polytrichopsida may have independent origins¹⁹."

- p. 15. Explain a bit better what would be needed to show or refute parallel vs. homologous origins of nematodontous peristomes ("comparative ontogenetic and transcriptomes" is a bit vague).

Response: we reworded to: "However, secondary peristome loss is common throughout the diversification of mosses⁶ and a unique origin of the nematodontous peristome should not be rejected until further comparative ontogenetic and transcriptomic studies elucidate the genetic networks underlying the development of the nemato- and arthrodontous peristomes."

- p. 16. How about stating a "small" plurality of trees (16 vs. 11 does not seem very different to me).

Response: the word was added.

- p. 16. One scenario is listed about peristome loss at the end of this paragraph. For balance, please spell out the alternative briefly ("otherwise...").

Response: We reworded the sentence to: "Unless secondary peristome loss is invoked in *Oedipodium*, the two main peristomial architectures of mosses may have independent origins, otherwise a unique origin of the peristome in mosses remains viable."

- p. 17. The "Buxbaumiidae... Diphysciidae..." consecutive branching has been confirmed in other studies (e.g., summary in Fig. 2 of Chang et al. 2014) – how about noting that?

Response: We added a reference (i.e., 16): "Our inferences confirm^{16, 21} that within the Bryopsida, the earliest splits gave rise to the Buxbaumiidae, and then the Diphysciidae (Fig. 1, and Supplementary Figs. 7–25)."

And the "Funariidae... Timmiidae..." consecutive divergences (particularly Timmiidae placement) were I think nailed down first by Chang et al. (2014) in terms of being well supported, so also mention that on p. 18?

Response: We added a reference to this previous study: "Consequently, the phylogenetic significance of the asymmetric cell division is ambiguous. However, the consecutive divergence of the Funariidae and Timmiidae (**first proposed by Chang et al. ¹⁶**) ...".

- p. 17. The 71 kb inversion is just a synapomorphy of a subset of Funariales, and so the inversion lacking Gigaspermales could still be the sister group of that group. In other words, this is an agnostic character when it comes to the possible existence of a clade comprising the two (it doesn't confirm or deny the possibility).

Response: We clarified the sentence: "The first evidence against a uniquely shared ancestry between the Gigaspermaceae and the remainder of the Funariales consisted in a large (71 kb) inversion in the plastid genome of the Funariaceae and Disceliaceae and also of the Encalyptales but lacking in the Gigaspermaceae, which were therefore accommodated in their own order, Gigaspermales⁵⁹."

- p. 19. It is disappointing that some major relationships (Splachnales etc) are still ambiguous here, but hyb-seq won't solve all problems. Perhaps discuss what might be needed?

Response: We reworded the sentence to: "Enhancing taxon sampling and extending character sampling to more exons and possibly their introns may be needed to finally resolve these relationships."

- p. 19. I am a bit uncomfortable with phrases like "the grade... leading to the Hypnanae" which make it sound like the latter are a pinnacle of evolution (and by implication other modern descendants are not). The point is that grades don't "lead" anywhere – they are artificial

constructs of our taxonomic systems (clades are natural, not artificial, in contrast). Please avoid such “progressivist” language.

Response: We replaced the words “leading to” by “subtending”, and the sentence now reads: “The Orthotrichales, Orthodontiales, and Aulacomniales complete the grade of largely acrocarpous Bryidae subtending the Hypnanae that is supported by concatenated nuclear data and a plurality of loci (Fig. 1). “This wording was also modified in other sentences, i.e., p. 4 & 14.

- p. 28. Where only these two AA substitution models considered??

Response: Yes, as these are two recently published empirical substitution models specific to green plant plastid and mitochondrial data, and fit better than any other “older” more widely used models such as cpREV, LG, WAG etc.

- p. 26. Did you do any “spot inspections” and adjustments of alignments, or just completely trust the automated aligners?

Response: we did not check the gene alignments, but a gene tree was manually inspected for every alignment, which would have revealed large problems in alignment (very long branches). We believe the in-frame alignments of genes are sufficiently robust.

- Figure S4. Would you expect high error rates on the genes with lowest coverage here? Is it a problem here?

Response: By “coverage” on this figure we are referring to the percentage of genes recovered for a particular sample. We now realize this is easily confused with another definition used in NGS studies—depth of sequencing. We have changed the y-axis in Figure S4 for clarity. Regarding sequencing depth, in Figure S4 we show that the average depth across genes for most specimens is over 50x, and no specimens have an average depth of less than 20x. We believe this to be sufficient sequencing depth to avoid systemic sequencing errors.

- Figure S4. Are any of the “low coverage” taxa for nuclear genes at critical points in the phylogeny? There is nice simulation work by Wiens and others suggesting that sparsely occupied alignments are not necessarily a problem for phylogenetic inferences (perhaps cite this literature in reference to this?)

Response: Yes, the figure shows that recovery of the nuclear genes decreased the further the phylogenetic distance from the taxa used to design the probes. This includes non-Bryopsida mosses including Takakia, Sphagnum, and Andreaea. However, this did not affect our ability to infer relationships on these nodes, because we supplemented the HybSeq data with sequences taken from 1KP data for the same genes. In each case where both HybSeq and transcriptome data were used, sequences from the same taxon were monophyletic. We also include a sentence toward the end of the section stating: “However, missing data, which tend to be

common in phylogenomic datasets, may not have significant effects on phylogenetic inferences” with a reference to two studies by Wiens.

Tables and figures

- Table S3. LWL = ?

Response: We noted below Table S3 that LWL means K_a and that k_s were measured using the LWL85 model reported by Drouin et al. 2008.

- Figs. S1-S3. Cell sizes seem uneven vertically (some lines not visible?) and taxon labels seem a bit offset from rows.

Response: We adjusted the spacing for gene and taxon labels in Figs S1-S3.

- Fig. S3. X-axis labels with gene names. Provide guiding text in the table header on how to look up what these are.

Response: The axis is labelled with numbers that correspond to the albeit arbitrary identifiers of the genes; these identifiers are used in the alignment files, the probe sequence files, and the target file for HybPiper.

General wording issues:

- use ‘plastid’ (the general organelle) not ‘chloroplast’ (one kind of plastid)

Response: we correct the one use of the word chloroplast on page 17.

- The phrase ‘Clade X composes Y’ is used many times in the MS. This is an awkward construction in English. More usually it would be written “clade X is composed of...” or “clade C comprises...”

Response: the verb “compose” can be used as a synonym of “make up”. In all sentences the reviewer may be referring to, the word is used in that sense: parts making up a larger item.

- Andreaeobryophytina and Andreaeophytina consistently compose a robust monophyletic group...
- The typically peristomate mosses, or Bryophytina, compose a robust lineage...
- Oedipodiopsida, Polytrichopsida and Tetrarhizopsida (hereafter OPT), compose a grade ...
- Tetrarhizopsida and Polytrichopsida compose the sister lineage to the Oedipodiopsida ...
- The remainder of this subclass composes a highly robust and concordant clade ...
- Aulacomniales compose the sister group to the Hypnanae...
- Hypopterygiaceae compose a lineage distinct from the other orders

We proceeded with changing the wording in one case on page 21: It is consistently shown here to **be** the sister taxon to the Ptychomniaceae.

- There are other constructions that need careful attention (often I think too much is crammed in one sentence and clauses could be broken up), such as the first sentence of the Abstract (very odd), and the sentence starting “These transformations...” on p. 2. There are other examples that should be identified and tackled by a skilled English writer.

Response: We did scan the entire for long sentence, and broke those down.

- Please use “amino acid” rather than “peptide” for AA alignments or substitution models (the former is much more commonly used, and is actually used in some of the Suppl. Info. (e.g., S24 legend)).

Response: changed throughout the text.

Specific wording issues:

- p. 2. ‘maintaining sexual reproduction’ or ‘allowing’ it?

Response: changed to “allowing for concurrent sexual reproduction”.

- p. 4. Avoid ‘since’ for non-temporal comparisons:

Response: one change made on page 4

- p. 5. ‘Persistent’ not ‘persisting’

Response: changed

- p. 10. Do you mean ‘non-overlapping genes with earlier studies’?

Response: changed as suggested

- p. 12. Define ‘organellar leakage’:

Response: clarified by rewording to: Organellar leakage, or relaxed organellar inheritance, ...

- p. 14. “no conflicting signal” ... conflicts... versus what?

Response: deleted the words “no conflicting signal”

- p. 14. I am not sure what “unique ancestry” means (monophyly?):

Response: reworded to “the monophyly of the OPT...”

- p. 15. Avoid archaic “whereby”

Response: The sentence was reworded.

- p. 15. Should this be “The monophyly of a clade comprising Tetrastichidae... etc”

Response: reworded as suggested.

- p. 18. “ambiguous” (line 1) – does this mean “poorly supported”?

Response: reworded as suggested.

- p. 19. “A highly robust and concordant clade” do you mean the clade itself (one branch to consider, the one supporting the clade... so what’s it concordant to?) or structure within the clade (if so, spell this out clearly).

Response: reworded to: The remainder of this subclass composes a highly robust clade, consistent among a plurality of gene trees, with the first splits segregating the Bryales and then the Rhizogoniales (Fig. 1).

- p. 25. “and then a similarity criterion...” – Words seem to be missing here.

Response: no, wording is complete and correct, as the text reads: “In such case, HybPiper uses a two-step strategy to choose among multiple full-length contigs: first a sequencing coverage depth cutoff — ..., and then a similarity criterion ...

- p. 29. Remove left parenthesis and space before “highlighted” on first line?

Response: Done.

- Ref. 59. Italics needed. Response: reference reads: Goffinet B, Wickett NJ, Werner O, Ros RM, Shaw AJ, Cox CJ. Distribution and phylogenetic significance of the 71-kb inversion in the plastid genome in Funariidae (Bryophyta). *Annals of Botany* **99**, 747–753 (2007). We do not see what the reviewer was referring to.

- p. 57. Why is “moss backbone” capitalized? Some phylogeneticists are uncomfortable with “backbone” (here and title) as the whole tree is a bifurcating system, not a caterpillar-spine (e.g., the “spine” shown “leading” to major clades – those we chose to focus on -- in the figures the “spine” is just one of many possible trackings of bifurcating branches along a tree, from root to tip).

Response: we changed the word “backbone” to “ordinal” throughout the text and supplementary documents.

Reviewer #2 (Remarks to the Author):

This by far the most robust moss phylogeny to date. This had been a major vexed question dating back to the last century. Though some issues still remain, this paper is likely to become the standard reference for some years to come. It contains very good major novel data sets and very good taxon sampling. The discussion is well balanced and extremely well argued.

All this is fine if you are well acquainted with moss diversity, as I am as the reviewer, but it’s very dense heavy going if you are not. To make this article more readable and of wider appeal I feel that the authors should consider broadening the discussion to include mention of the phylogenies of other groups (plant and animal) comparable mosses.

Response: We have added a paragraph at the beginning of the result and discussion section to provide a broader comparative context to our approach and study. We have also provided a

concluding paragraph that summarizes the outcome within a broader methodological context, and the main contributions of our study.

As it stands the focus is perhaps too narrow for Nature Communications. Minor points

Abstract. There may be a strict word limit here but it would be better if the key findings were included.

Response: the abstract refers to the main outcomes pertaining to relative substitution rates but does indeed only refer in general terms to the phylogenetic outcomes. Including explicit statements of the main results would significantly increase the length of the abstract, and hence lead us to exceed the number of allowed words.

Introduction change wording:-

The vegetative body has undergone developmental shifts to optimize vegetative growth while maintaining sexual reproduction. Teleology change shifts to optimize to shifts that

Response: changed as suggested

mosses arose from repeated burst of diversification - typo bursts Response: Corrected that may be triggered by extrinsic factors such as changes in global climate -wrong tense --may have been triggered

Response: Corrected.

Reviewer #1 (Remarks to the Author):

Re-review of “Resolution of the ordinal phylogeny of mosses using targeted exons from organellar and nuclear genomes” by Y. Liu et al.

I have read through the revised MS and author’s rebuttal to reviewer comments and think they have addressed essentially all the points raised (I have noted a few minor things below, which I am confident can be readily fixed without further review).

Overall, this is an important study for Nature Communications because it addresses the phylogenetic history of mosses, a key land-plant lineage that is important both for planetary ecology and for understanding land-plant evolution and diversity. Also, one of our key model organisms (*Physcomitrella*) is a moss, so this paper should have a broad audience. I urge Nature Communications to publish it.

The rebuttals to reviewer comments are all essentially fine , but I note a few minor things here:

(1) Original comment: - p. 9, 10. I think the non-bryophyte rates listed in parentheses here come from published studies, but I am not 100% sure (footnote to table S3?). Can this be made really clearer in the main text?

Author response: We do refer in the text on p. 9 to Palmer²⁹ and Drouin et al.³⁰ (the two main studies on this topic) and to Drouin et al.³⁰ in Table S3 where we report the rates they published. It is not clear to us how else this would be clarified.

ADDITIONAL COMMENT: I think I was confused by the word order in this footnote. How about something like: “Measurements are from Drouin et al. (6), based on the LWL85 model” -- otherwise it looks like the citation is for the model, not the measurements?

(2) The authors have done an excellent job of looking at different methods and assumptions. While I firmly agree that no more analyses are needed here given the substantial effort already put into this, it would also be very straightforward to add a single statement/caveat that further investigations of models and partitioning schemes could provide additional insights into other possible sources of bias in analysis. This is partly borne out by research on other groups, so it is not entirely speculative, and possibly not of minor importance.

(3) Original comment: ... The phrase 'Clade X composes Y' is used many times in the MS. This is an awkward construction in English. More usually it would be written "clade X is composed of..." or "clade C comprises..."

Response: the verb "compose" can be used as a synonym of "make up". In all sentences the reviewer may be referring to, the word is used in that sense: parts making up a larger item.

- Andraeobryophytina and Andraeophytina consistently compose a robust monophyletic group...
- The typically peristomate mosses, or Bryophytina, compose a robust lineage... • Oedipodiopsida, Polytrichopsida and Tetrarhizopsida (hereafter OPT), compose a grade
- ... • Tetrarhizopsida and Polytrichopsida compose the sister lineage to the Oedipodiopsida
- ... • The remainder of this subclass composes a highly robust and concordant clade ... • Aulacomniales compose the sister group to the Hypnanae... • Hypopterygiaceae compose a lineage distinct from the other orders

ADDITIONAL COMMENT: Sorry, but this usage of compose is non-idiomatic (and the repeated use strikes a really odd note to my ear). How about replacing at least a few of these instances with a different verb like "are" or "represents" for some diversity in style?

(3) Original comment: - p. 25. "and then a similarity criterion..." – Words seem to be missing here.

Response: no, wording is complete and correct, as the text reads: "In such case, HybPiper uses a two-step strategy to choose among multiple full-length contigs: first a sequencing coverage depth cutoff — ..., and then a similarity criterion .

ADDITIONAL COMMENT: (Now page 28). Ah...!! I think the long-dash is being used instead of parentheses, so you should move the second long-dash up to just before "and then a similarity criterion", right?

(4) Original comment: - - Ref. 59. Italics needed.

Response: reference reads: Goffinet B, Wickett NJ, Werner O, Ros RM, Shaw AJ, Cox CJ. Distribution and phylogenetic significance of the 71-kb inversion in the plastid genome in Funariidae (Bryophyta). *Annals of Botany* 99, 747–753 (2007). We do not see what the reviewer was referring to.

ADDITIONAL COMMENT: Sorry, I meant ref. 58 not 59 as it was (now ref. 71): Basically "Oedipodium" in Ligrone & Duckett (2018) should be italicized.

Additional comments on this version:

p. 3. How about “that make up the ordinal relationships of...”, and (p. 6) “... resolve the ordinal-level phylogeny...”

p. 7. Remove comma after “uncommon” (third line from end)

p. 11. Lines 5 and 7, the numbers in brackets are given in the order in the cited table, but are not the order of corresponding items in the preceding text--please re-write, this confused me. I think you could either rearrange the sentence items (lines 1-3: plastid... nuclear... mitochondrial) to match the number order (1, 5.7, 7.4) or vice versa. And the same for the numbers on line 5.

p. 15, ‘Distinctive ontogeny’ not “distinct ontogeny”.

p. 16. Sometimes use OPT, sometimes OTP. Be consistent or explain if there is a distinction.

References: I am also not sure of the use of “ed[^]” in several references (refs. 6, 17, 18, 24, 60 and 69).

Reviewer #2 (Remarks to the Author):

This revised paper has addressed all the issues raised by the reviewers. It is by far the most authoritative moss phylogeny produced to date. This is a highly novel first class piece of work which will be widely read and will become a key reference.

A particular strength of the paper is its well balanced consideration of problematic issues.

Responses to comments by reviewers and requests by the editor (p. 3) on NCOMMS-18-23931B.

Responses to reviewers

Reviewer #1 (Remarks to the Author):

Re-review of “Resolution of the ordinal phylogeny of mosses using targeted exons from organellar and nuclear genomes” by Y. Liu et al.

I have read through the revised MS and author’s rebuttal to reviewer comments and think they have addressed essentially all the points raised (I have noted a few minor things below, which I am confident can be readily fixed without further review).

Overall, this is an important study for Nature Communications because it addresses the phylogenetic history of mosses, a key land-plant lineage that is important both for planetary ecology and for understanding land-plant evolution and diversity. Also, one of our key model organisms (*Physcomitrella*) is a moss, so this paper should have a broad audience. I urge Nature Communications to publish it.

The rebuttals to reviewer comments are all essentially fine, but I note a few minor things here:

(1) Original comment: - p. 9, 10. I think the non-bryophyte rates listed in parentheses here come from published studies, but I am not 100% sure (footnote to table S3?). Can this be made really clearer in the main text?

Original Author response: We do refer in the text on p. 9 to Palmer²⁹ and Drouin et al.³⁰ (the two main studies on this topic) and to Drouin et al.³⁰ in Table S3 where we report the rates they published. It is not clear to us how else this would be clarified.

ADDITIONAL COMMENT: I think I was confused by the word order in this footnote. How about something like: “Measurements are from Drouin et al. (6), based on the LWL85 model” -- otherwise it looks like the citation is for the model, not the measurements?

Response to additional comment: we have changed the sentence in the footnote of Table S3 as proposed by the reviewer.

(2) The authors have done an excellent job of looking at different methods and assumptions. While I firmly agree that no more analyses are needed here given the substantial effort already put into this, it would also be very straightforward to add a single statement/caveat that further investigations of models and partitioning schemes could provide additional insights into other possible sources of bias in analysis. This is partly borne out by research on other groups, so it is not entirely speculative, and possibly not of minor importance.

(3) Original comment: ... The phrase ‘Clade X composes Y’ is used many times in the MS. This is an awkward construction in English. More usually it would be written “clade X is composed of...” or “clade C comprises...”

Original Response: the verb “compose” can be used as a synonym of “make up”. In all sentences the reviewer may be referring to, the word is used in that sense: parts making up a larger item.

- Andreaeobryophytina and Andreaeophytina consistently compose a robust monophyletic group...
- The typically peristomate mosses, or Bryophytina, compose a robust lineage... • Oedipodiopsida, Polytrichopsida and Tetrarhizopsida (hereafter OPT), compose a grade ... • Tetrarhizopsida and Polytrichopsida compose the sister lineage to the Oedipodiopsida ... • The remainder of this subclass composes a highly robust and concordant clade ... • Aulacomniales compose the sister group to the Hypnanae... • Hypopterygiaceae compose a lineage distinct from the other orders

ADDITIONAL COMMENT: Sorry, but this usage of compose is non-idiomatic (and the repeated use strikes a really odd note to my ear). How about replacing at least a few of these instances with a different verb like “are” or “represents” for some diversity in style?

Response to additional comment: we have replaced “compose” in most instance throughout the text.

(3) Original comment: - p. 25. “and then a similarity criterion...” – Words seem to be missing here.

Original Response: no, wording is complete and correct, as the text reads: “In such case, HybPiper uses a two-step strategy to choose among multiple full-length contigs: first a sequencing coverage depth cutoff — ..., and then a similarity criterion.

ADDITIONAL COMMENT: (Now page 28). Ah...!! I think the long-dash is being used instead of parentheses, so you should move the second long-dash up to just before “and then a similarity criterion”, right?

Response to additional comment: we have inserted a long dash.

(4) Original comment: - - Ref. 59. Italics needed.

Original Response: reference reads: Goffinet B, Wickett NJ, Werner O, Ros RM, Shaw AJ, Cox CJ. Distribution and phylogenetic significance of the 71-kb inversion in the plastid genome in Funariidae (Bryophyta). *Annals of Botany* 99, 747–753 (2007). We do not see what the reviewer was referring to.

ADDITIONAL COMMENT: Sorry, I meant ref. 58 not 59 as it was (now ref. 71): Basically “*Oedipodium*” in Ligrone & Duckett (2018) should be italicized.

Response to additional comment: *Oedipodium* is now italicized

p. 3. How about “that make up the ordinal relationships of...”, and (p. 6) “... resolve the ordinal-level phylogeny...”

Response to additional comment: we have added words as proposed in each case.

p. 7. Remove comma after “uncommon” (third line from end)

Response to additional comment: we removed the comma.

p. 11. Lines 5 and 7, the numbers in brackets are given in the order in the cited table, but are not the order of corresponding items in the preceding text--please re-write, this confused me. I think you could either rearrange the sentence items (lines 1-3: plastid... nuclear... mitochondrial) to match the number order (1, 5.7, 7.4) or vice versa. And the same for the numbers on line 5.

p. 15, ‘Distinctive ontogeny’ not “distinct ontogeny”.

Response to additional comment: we changed the word as proposed.

p. 16. Sometimes use OPT, sometimes OTP. Be consistent or explain if there is a distinction.

Response to additional comment: we now use OPT consistently. Thank you for noticing!

References: I am also not sure of the use of “ed^” in several references (refs. 6, 17, 18, 24, 60 and 69).

Response to additional comment: we have corrected these.

Reviewer #2 (Remarks to the Author):

This revised paper has addressed all the issues raised by the reviewers. It is by far the most authoritative moss phylogeny produced to date. This is a highly novel first class piece of work which will be widely read and will become a key reference.

A particular strength of the paper is its well balanced consideration of problematic issues.